# Dietary Patterns and Prevalent NAFLD at Year 25 from the Coronary Artery Risk Development in Young Adults (CARDIA) Study

**DOI:** 10.3390/nu14040854

**Published:** 2022-02-18

**Authors:** Meagan E. Gray, Sejong Bae, Rekha Ramachandran, Nicholas Baldwin, Lisa B. VanWagner, David R. Jacobs, James G. Terry, James M. Shikany

**Affiliations:** 1Division of Gastroenterology and Hepatology, University of Alabama at Birmingham, Birmingham, AL 35294, USA; 2Division of Preventive Medicine, University of Alabama at Birmingham, Birmingham, AL 35294, USA; sbae@uabmc.edu (S.B.); rramachandran@uabmc.edu (R.R.); jshikany@uabmc.edu (J.M.S.); 3Department of Internal Medicine, University of Alabama at Birmingham, Birmingham, AL 35294, USA; nbaldwin@uabmc.edu; 4Department of Medicine, Division of Gastroenterology and Hepatology, Northwestern University, Evanston, IL 60208, USA; lvw@northwestern.edu; 5Department of Preventive Medicine, Division of Epidemiology, Northwestern University, Evanston, IL 60208, USA; 6School of Public Health, University of Minnesota, Minneapolis, MN 55455, USA; jacob004@umn.edu; 7Department of Radiology and Radiological Sciences, Vanderbilt University Medical Center, Nashville, TN 37232, USA; james.g.terry@vumc.org

**Keywords:** nonalcoholic fatty liver disease, dietary pattern, diet quality

## Abstract

The prevalence of nonalcoholic fatty liver disease is rapidly rising. We aimed to investigate associations of diet quality and dietary patterns with nonalcoholic fatty liver disease (NAFLD) in Black and White adults. We included 1726 participants who attended the Year 20 Exam of the Coronary Artery Risk Development in Young Adults (CARDIA) study and had their liver attenuation (LA) measured using computed tomography at Year 25 (2010–2011). NAFLD was defined as an LA of ≤51 Hounsfield units after the exclusion of other causes of liver fat. The a priori diet-quality score (APDQS) was used to assess diet quality, and dietary patterns were derived from principal components analysis. Univariate and multivariable logistic regression models were used to evaluate the association between the APDQS, dietary patterns, and NAFLD, and were adjusted for Year 20 covariates. NAFLD prevalence at Year 25 was 23.6%. In a model adjusted for age, race, sex, education, alcohol use, physical activity, smoking, and center at Year 25, the APDQS was inversely associated (*p* = 0.004) and meat dietary pattern was positively associated (*p* < 0.0001) with NAFLD, while the fruit-vegetable dietary pattern was not significantly associated (*p* = 0.40). These associations remained significant when additionally adjusting for comorbidities (type 2 diabetes mellitus, dyslipidemia, hypertension), however, significant associations were diminished after additionally adjusting for body mass index (BMI). Overall, this study finds that the APDQS and meat dietary patterns are associated with prevalent NAFLD in mid-life. The associations appear to be partially mediated through higher BMI.

## 1. Introduction

Nonalcoholic fatty liver disease (NAFLD) is the most common cause of chronic liver disease worldwide, affecting approximately 25% of the adult population globally [1] and 31% (83.1 million) of adults in the United States (US) [2]. Thus far, there remains a lack of effective pharmacologic treatment; thus, the need for population-based preventive strategies is critical.

Specific macronutrients and foods have been implicated in the development of NAFLD, specifically saturated fat, refined carbohydrates, and red and processed meats, and these have been previously reviewed in detail [3]. Several studies have evaluated diet quality and NAFLD, with findings consistently showing an inverse relationship between diet quality and risk of NAFLD [4,5,6,7], however, few prior studies have the imaging, thorough characterization, and phenotyping of participants for accurate exposure and outcome definitions. The most significant variation in prior studies is the definition of NAFLD used, which ranges from Medicare claims (which may miss a significant number of affected patients), to noninvasive laboratory-based scoring systems (i.e., the Fatty Liver Index (FLI)), to robust magnetic resonance spectroscopy. NAFLD is most accurately defined as the presence of > 5% hepatic steatosis either on imaging or liver histology after the exclusion of secondary causes of fat accumulation (i.e., significant alcohol consumption, certain medical conditions, and specific medications). Ruling out secondary causes of hepatic steatosis can be challenging in retrospective and electronic health record cohorts. Diet assessment methodology is also widely variable. 

Only a few studies have evaluated associations of dietary patterns, which may more meaningfully reflect long-term eating habits, and risk of NAFLD [8,9,10,11,12], however, the patterns described are also highly variable. The only study evaluating dietary patterns in US adults defined two patterns a posteriori, one of “vitamins, minerals, and fiber” and the second “high levels of saturated and mono-unsaturated fatty acids, total fat and carbohydrates” [12]. These patterns may not correlate with real-life eating practices. The authors also used the FLI to define NAFLD without imaging or histologic assessment. 

The Coronary Artery Risk Development in Young Adults (CARDIA) study provides a unique opportunity to investigate diet and risk of prevalent NAFLD at midlife in US adults utilizing computed tomography (CT) imaging to diagnose NAFLD. CARDIA is one of the longest running studies of its kind, enrolling over 5000 Black and White adults at baseline, and including participants at 4 major population centers across the US. Robust prospective data collection in addition to CT imaging also allows for an enhanced assessment of hepatic steatosis in order to more accurately define the population with NAFLD. In this report, we present associations of food groups, a priori diet quality score, and two different dietary patterns-meat and fruit-vegetable–with risk of NAFLD. We hypothesized that the a priori diet quality score and fruit-vegetable dietary pattern would be inversely associated with NAFLD, whereas the meat pattern would be positively associated with NAFLD. 

## 2. Materials and Methods

### 2.1. Study Design and Sample

CARDIA is a multicenter, prospective, longitudinal cohort study of lifestyle and cardiovascular risk factors in young adults. Between 1985 and 1986, 5115 participants were enrolled across 4 U.S. cities (Birmingham, AL, USA; Chicago, IL, USA; Minneapolis, MN, USA; Oakland, CA, USA). The study design has been previously published [13]. Participants were balanced by sex, race (White or Black), age (18–24 or 25–30 years old), and education level (≤high school or >high school). Follow-up visits were conducted at years 2, 5, 7, 10, 15, and 25, with retention of 72% of participants at Year 25 (2010–2011) and the study is ongoing. The study protocol was approved by institutional review boards at each participating institution, and written informed consent was obtained from participants at each follow-up examination.

The present report includes 1726 CARDIA participants who attended the Year 20 and Year 25 Exam and who underwent cross-sectional imaging with non-contrast CT of the abdomen at Year 25 (Figure 1). Participants with missing covariates were excluded in addition to those with missing diet variables, implausible calorie counts (women with daily kilocalories <600 or >6000, and men with daily kilocalories <800 or >8000), and those with alternative causes for hepatic steatosis (heavy alcohol use, human immunodeficiency virus, chronic hepatitis C virus, steatogenic medications, and missing variables to determine alternative cause). Possible confounding variables, including demographic, clinical, and others, were selected based on literature review and known clinical risk factors. These variables were adjusted for in the multiple logistic regression models. 

### 2.2. Data Collection

Data collection was standardized across all field centers and has been previously described [13]. Demographic and clinical characteristics were collected of the participants present at each exam. For this study, we used data collected at exam Year 20 for covariates. These variables included age, body mass index (BMI), race, sex, income, education level, alcohol use, smoking, physical activity, blood pressure, comorbidities, medications, field center, and healthcare access.

Body weight was measured with a calibrated balance beam to the nearest 0.2 kg. Height was measured to the nearest 0.5 cm with a vertical ruler. Blood pressure was measured in a seated position three times at 1-min intervals after 5 min of resting, and the latter 2 measurements were averaged for use in the analysis. Blood was drawn in a fasted state and seated position. After separation, frozen plasma (−70 °C) was shipped for analysis in a central laboratory [13]. Glucose was measured using the hexokinase method and insulin by the Elecsys sandwich immunoassay [14,15]. The cholesterol panel (total cholesterol, high-density lipoprotein (HDL) cholesterol, and triglycerides) was measured at a central laboratory [16] and low-density lipoprotein (LDL) cholesterol was calculated using the Friedewald equation [17,18]. 

Obesity was defined as a BMI ≥ 30 kg/m^2^, hypercholesterolemia as a total cholesterol ≥240 mg/dL or use of lipid-lowering medication, and hypertension as systolic blood pressure ≥140 mmHg, diastolic blood pressure ≥90 mmHg, and/or antihypertensive medication use. Diabetes was defined as fasting plasma glucose ≥126 mg/dL, 2-h post-challenge glucose ≥200 mg/dL, hemoglobin A1C ≥6.5%, and/or treatment with insulin or hypoglycemic agent. The modified National Cholesterol Education Program Adult Treatment Panel III criteria defined the metabolic syndrome [19]. Physical activity was quantified as exercise units (EU) and was assessed with the CARDIA physical activity questionnaire [20].

### 2.3. Diet Measure and Dietary Pattern Scores

At the Year 20 Exam, dietary intake was assessed using the CARDIA diet history obtained by a trained interviewer. Food and beverages consumed were assigned to one of 166 created food groups devised by the Nutrition Coordinating Center (NCC) at the University of Minnesota (Minneapolis, MN, USA) and based on a modified US Department of Agriculture (USDA) food grouping system. Servings were reported using USDA recommendations. Individual food-group intake was calculated as the total number of standard servings reported per day of each food within a given food group. The 166 NCC food groups were collapsed into 46 food groups, as in previous CARDIA analyses [21].

The diet quality score was defined a priori and was previously created in CARDIA [21]. Food groups are classified as beneficial (20 groups), adverse (13 groups), or neutral (13 groups) based on their association with disease. The beneficial and adverse groups were broken down into quintiles of consumption and participants assigned a score from 0 to 4, depending on their level of consumption. In groups with large subsets of non-consumers, non-consumers were coded as 0 and consumers were divided into quartiles (scores 1 to 4). The a priori diet quality score (APDQS) was the sum of category scores 0–4 for beneficial food groups plus scores in reverse order (4–0) for adverse food groups, with a maximum APDQS of 132. Neutral food groups were not included in the calculation.

Principal components analysis (PCA) with orthogonal rotation was used to derive uncorrelated dietary patterns from the 46 food groups, as described in a previous CARDIA study [21]. Based on this PCA, two major dietary patterns were derived; the meat- and fruit-vegetable dietary patterns, which reflect their relative high amounts of meat and fruit/vegetables, respectively.

### 2.4. Assessment of Hepatic Steatosis

Hepatic steatosis was assessed using non-contrast CT of the abdomen obtained at the Year 25 Exam. The CT scans were performed using multidetector CT scanners from either General Electric (GE 750HD 64 and GE LightSpeed VCT 64, Birmingham and Oakland Field Centers, respectively; GE Healthcare, Waukesha, WI, USA) or Siemens (Sensation 64, Chicago and Minneapolis Field Centers; Siemens Medical Solutions, Erlangen, Germany). The CT diagnosis of hepatic steatosis was made by measuring liver attenuation (LA) in Hounsfield units (HU) [22,23]. The measured LA decreases as the amount of hepatic fat increases (low LA = high hepatic fat) [22]. LA was measured using the right lobe of the liver. Quantitative measurements were performed using a dedicated workflow within the National Institute of Health’s Center of Information Technology Medical Image Processing, Analysis, and Visualization (MIPAV) application (http://mipav.cit.nih.gov/index.php, accessed on 7 January 2022). The LA was determined by averaging nine measurements on three slices using circular regions of interest of 2.6 cm^2^ avoiding large vessels and liver lesions. High reproducibility of CT measured LA has been previously shown [24]. 

It has previously been shown that an LA value of ≤40 HU on unenhanced CT correlates with >30% (moderate/severe) steatosis [23,25,26,27]. Liver attenuation values ≤51 HU on unenhanced CT indicate at least mild steatosis and can be used to define NAFLD in the absence of other causes of liver fat (e.g., alcohol use, medications, intravenous drug use, viral infection) [23,25,26,27]. Therefore, we defined any NAFLD as LA of ≤51 HU after exclusion of other causes of liver fat. Participants with heavy alcohol use (defined as >14 standard drinks per week for women or >21 standard drinks per week for men), human immunodeficiency virus (HIV), hepatitis C virus (HCV), and medications known to cause hepatic steatosis (e.g., amiodarone, diltiazem, methotrexate, valproate, tamoxifen), were excluded (Figure 1).

### 2.5. Statistical Analysis

Frequencies (percent) and means (±standard deviation) were used to summarize categorical and continuous variables, respectively, by NAFLD status. Student’s *t*-test and Chi-square tests were used to compare categorical and continuous variables, respectively, between NAFLD status. Univariate and multivariable logistic regression models were used to evaluate the association between the a priori diet quality score (APDQS), dietary patterns and NAFLD, adjusting for relevant covariates, including age, sex, income, education level, estimated glomerular filtration rate (GFR), medications, alcohol use, smoking, level of physical activity, medical comorbidities (hypertension, hyperlipidemia, type 2 diabetes mellitus, metabolic syndrome), field center, and variables related to access to healthcare. Medical comorbidities and BMI were felt to be likely mediators and were examined as covariates, one at a time. Given the findings, the comorbidities were then grouped into a single adjusted model for parsimonious reporting. The models estimated the unadjusted measures of association, as well as multivariable adjusted models with pre-defined contrasts among quartiles. Potential multicollinearity of predictor variables in models was assessed using the variance inflation factor, with a threshold of five. Analyses were conducted by SAS version 9.4 (SAS Institute, Inc., Cary, NC, USA). A *p*-value ≤ 0.05 was used for statistical significance.

## 3. Results

Out of the 3498 participants that attended the Year 25 exam, there were 1726 (49.3%) eligible participants included in the analysis, of which 408 (23.6%) had NAFLD. The demographics and clinical characteristics by NAFLD status are summarized in Table 1, and comparison of included and excluded participants can be reviewed in Appendix A. The proportion of NAFLD was higher in males. While those with NAFLD were slightly less educated, there was no difference in income or access to medical care. Participants with NAFLD had significantly higher BMI, prevalence of obesity (BMI ≥ 30 kg/m^2^), and waist circumference, as well as a higher prevalence of metabolic comorbidities, including hyperlipidemia, hypertension, diabetes mellitus, and the metabolic syndrome. Participants with NAFLD were also more likely to be current cigarette smokers and reported less physical activity. 

The unadjusted associations between energy intake and individual food groups with NAFLD are shown in Table 2. Participants with NAFLD had a significantly higher mean daily energy intake compared to those without. The consumption of potatoes, grains, meat and fish, dairy, fats, and beverages were significantly higher among those with NAFLD while those without NAFLD consumed more fruit. There was no difference between the groups in intake of other food groups, specifically vegetables (excluding potatoes); beans; eggs/omelets; seeds, nuts, peanut butter; salad dressing/sauces; soy/nondairy foods; pickled foods; chocolate; sweet extras; or sugar substitutes.

Logistic regression was performed to model NAFLD at Year 25 as a function of APDQS and meat- and fruit-vegetable dietary patterns at Year 20 (Table 3). In a model adjusted for age, race, sex, education, alcohol use, physical activity, smoking, and field center at Year 25, the APDQS was inversely associated (Type 3 Chi-square *p* = 0.004) with NAFLD, with each increasing quartile having lower odds ratio (OR) for NAFLD compared to the lowest quartile (OR 0.53, 95% Confidence Interval (CI) (0.36–0.79) for Quartile 4 vs. 1). Using this same model, the meat dietary pattern was positively associated (Type 3 Chi-square *p* < 0.0001) with NAFLD, with increasing odds for NAFLD across each increasing quartile compared to the lowest quartile (OR 2.7, 95% CI (1.83–3.99) for Quartile 4 vs. 1), while the fruit-vegetable dietary pattern was not significantly associated (Type 3 Chi-square *p* = 0.40).

Our second model additionally adjusted for relevant comorbidities, felt to be possible mediators (type 2 diabetes mellitus (T2DM), hyperlipidemia, hypertension, and the metabolic syndrome) (Table 3, model 2). In this model, the ADPQS remained inversely associated (Type 3 Chi-square *p* = 0.04) with NAFLD, the meat dietary pattern remained positively associated (Type 3 Chi-square *p* = 0.014), and the fruit-vegetable dietary pattern remained insignificantly associated (Type 3 Chi-square *p* = 0.46). We suspected that BMI may be a significant contributor to the association between our diet quality score and dietary patterns and NAFLD, and thus added this additional likely mediator into our fully adjusted analysis last (Table 3, model 3). With the addition of BMI, our APDQS and meat dietary pattern lost significant independent associations with prevalent NAFLD (Type 3 Chi-square *p* = 0.11 and Type 3 Chi-square *p* = 0.26, respectively) and the fruit-vegetable pattern remained not significantly associated (Type 3 Chi-square *p* = 0.16). A visual representation of NAFLD proportions by diet scores is shown in Figure 2. 

## 4. Discussion

In the present study, we evaluated the associations of the APDQS and two dietary patterns (meat- and fruit-vegetable) with risk of NAFLD in participants in the CARDIA study. Overall, we demonstrated that that the associations between both the APDQS and the meat dietary pattern and NAFLD were independent of circulating cardiometabolic risk factors, but associations are likely mediated through higher generalized obesity.

Our findings are similar to prior analyses, which have shown that poorer diet quality is associated with increased risk of NAFLD [4,5,7,28,29,30]. Three prior studies evaluated specifically diet quality and NAFLD in US adults. In a cross-sectional study of 1861 participants of the Multiethnic Cohort (MEC) study, the association of 4 diet quality scores (based on a food frequency questionnaire at study entry 1993–96) and the prevalence of NAFLD (defined at >5.5% liver fat on MRI) at Year 20 (2013–16) follow-up was evaluated. This cohort comprised adults ages 45–75 years from Hawaii and Los Angeles, California with almost equal proportions of Japanese American, Native Hawaiian, African American, White, and Latino participants. Participants were equally stratified across sex and BMI, with approximately one third of participants normal weight, one third overweight, and one third obese. Across all of the diet quality scores (Healthy Eating Index (HEI), Alternative HEI (AHEI), the Alternate Mediterranean Diet Score (AMDS), and the Dietary Approaches to Stop Hypertension (DASH)), which were defined a priori, higher dietary quality scores at study entry were inversely related to percent liver fat on a continuum (ß_s_ = −0.14 to −0.08) at Year 20 follow-up [5]. Approximately 33% of participants had NAFLD, and associations between diet quality scores and NAFLD prevalence were not described. 

A more recent nested case-control study from the MEC study, defining NAFLD and cirrhosis by Medicare claims, evaluated the association between 4 diet quality scores and prevalence of NAFLD [30]. There were 2959 NAFLD cases (509 with cirrhosis, 2450 without) and 29,292 matched controls. This population was older as it included those who reached age 65 years and were linked to Medicare services between 1999–2016 and included significantly more Japanese American (~50% of the population) and Latino (~20%) participants. Using the same 4 diet quality scores, obtained at cohort entry, higher HEI and DASH scores were inversely associated with NAFLD risk (OR: 0.83; 95% CI: 0.73, 0.94; *p*_trend_ = 0.002, and OR: 0.78; 95% CI: 0.69, 0.89; *p*_trend_ < 0.001, respectively). The inverse association was stronger for NAFLD with cirrhosis compared to NAFLD without cirrhosis. Given the high prevalence of NAFLD in Japanese American and Latino adults compared to White and Black adults, in addition to higher prevalence in older adults, effects of diet may be more pronounced in this population compared to our population.

The prevalence of NAFLD in association with diet quality has also been evaluated in the third National Health and Nutrition Examination Survey (NHANES III) from 1988–1994 [7]. The NHANES III survey sampled the adult, non-institutionalized population in the United States across ages 20–74 years and included 10,858 participants. The mean age was 42.9 years, and the population was mostly White (~75%). HEI was used to define diet quality, and ultrasound to define NAFLD. The prevalence of NAFLD across the quartiles of HEI scores ranged from 32.3 to 35.4% (*p* = 0.407), which was surprising given the BMI range of 26.3–26.9 kg/m^2^, diabetes prevalence of 6.8–7.2% and alanine aminotransferase (ALT) range of 17.2–17.8 IU/L across the quartiles. After adjustment, however, for age, sex, race, ethnicity, education level, economic status, BMI, smoking status, diabetes, hypertension, caffeine consumptions, ALT, alcohol consumption, high-density lipoprotein cholesterol, c-reactive protein, transferrin saturation, sedentary lifestyle, and total energy intake, the HEI became significantly and inversely associated to the prevalence of NAFLD in a dose-dependent manner (*p*_trend_ = 0.028). Our population differed from the NHANES population with more Black participants, and more participants with obesity, diabetes, and the metabolic syndrome. Given that Black adults have a higher prevalence of obesity and components of the metabolic syndrome, yet a lower prevalence of NAFLD, this may have affected the impact of the a priori diet quality score and dietary patterns that we assessed. 

Lastly, Ma et al. [4] evaluated 1521 participants from the Framingham Heart Study to determine if dietary patterns could affect the prevalence and incidence of NAFLD over 6 years of follow up. Utilizing CT images and the Mediterranean-style diet score (MDS) and AHEI, changes in diet quality and liver fat were determined between baseline and 6-year follow-up exams. A liver-phantom ratio (inversely related to liver fat) was used to define NAFLD. For each 1-standard deviation increase in the MDS and the AHEI from baseline to follow-up exam, the liver-phantom ratio increased (liver fat decreased) by 0.57 (95% CI, 0.27–0.86; *p* < 0.001) and 0.56 (95% CI, 0.29–0.84; *p* < 0.001), respectively, and the odds for incident fatty liver decreased by 26% (95% CI, 10–39%; *p* = 0.002) and 21% (95% CI, 5–35%; *p* = 0.02), respectively. Although observational, these findings suggested that improving dietary pattern may decrease prevalence and incidence of NAFLD. 

Our results also showed that the meat dietary pattern is associated with prevalent NAFLD in mid-life, but this significant association is likely mediated through BMI. The only study that derived its own dietary patterns to examine prevalence of NAFLD among US adults was published recently using the NHANES 2015–2016 database with 20,643 participants over age 18 years who were examined during the 2005–2012 NHANES cycles [12]. NAFLD was defined based on a noninvasive scoring system, the FLI. Participants in the highest quartile of a dietary pattern comprised of “vitamins, minerals, and fiber” had a 34% lower odds of prevalent fatty liver compared to the lowest quartile (OR 0.66; 95% CI 0.43–0.71), while those in the highest quartile of the dietary pattern comprised of “high levels of saturated and mono-unsaturated fatty acids, total fat and carbohydrates” had a higher OR for NAFLD compared to those in the lowest quartile (OR = 1.86; 95% CI 1.42–2.95). These dietary patterns may not be easily translatable to common eating behaviors in US adults and are contradictory to multiple prior studies showing the beneficial effects of mono-unsaturated fats on hepatic steatosis [31,32,33].

Our study is unique in the use of the a priori diet quality score, as well as the meat- and fruit-vegetable dietary patterns, which approximate Western and prudent diets, respectively. The APDQS is a useful score for diet quality in NAFLD as the classification of foods as beneficial, neutral, and adverse is in line with current data with regards to beneficial (fruits, vegetables, whole grains, lean meats, and unsaturated fats) and adverse (refined carbohydrates and saturated fats) foods for NAFLD [3]. Similarly, the meat dietary pattern would be expected to represent a diet high in saturated fat, known to be harmful in NAFLD. The association between individual food groups and NAFLD is in line with what other studies have shown, especially that meat, dairy, fat, grains, potatoes, and beverages are strongly positively associated with NAFLD. These foods would be expected to be comparable to that of a “Western” or “fast-food” dietary pattern evaluated in previous dietary pattern studies, which were also strongly predictive of NAFLD [10]. Notably, our analysis showed that fruit consumption was highest among those without NAFLD. This is an important point in an era where low-carbohydrate diets are often prescribed for patients with NAFLD. While refined carbohydrates, sweets, and sugary beverages worsen NAFLD [34,35], there are no data to suggest that whole grains or fruits are harmful. 

A focus on overall dietary scores and patterns is likely more relevant to actual dietary intake than individual nutrient associations with NAFLD; however, the current study refuted our hypothesis in the fully adjusted model that the APDQS and fruit-vegetable dietary pattern would be independently and inversely associated with NAFLD while the meat-pattern would be strongly associated. In Table 3 we show that the effects of the APDQS and meat dietary pattern are likely mediated through higher BMI, which is not surprising given that 71.6% of our NAFLD population was obese. Obesity is a well-known independent risk factor for NAFLD as shown in multiple prior studies [36,37,38], and participants followed prospectively in CARDIA with the largest increases in APDQS over 20 years had the lowest gains in BMI [39]. 

The current study has several strengths, including large sample size and nearly equal representation of men and women, Black and White participants, the use of an objective measure of hepatic steatosis, and comprehensive assessment of diet, including the assessment of the overall diet through a diet quality score and patterns. There are several limitations to this study. A significant number of participants who attended the Year 25 exam were excluded (*n* = 1772, 50.6%). There were no significant differences between age and gender of included versus excluded participants, however there was a significantly higher proportion of White participants in the included sample (59.7% vs. 46.7%, *p* < 0.0001) (Appendix A). The excluded participants had a higher BMI and a higher likelihood of metabolic disease than our included participants, which may have potentially reduced significance of our findings, however, the rigorous eligibility criteria likely produced a cleaner delineation of outcomes.

Another limitation involves the available imaging modality. Since quantitative assessment of steatosis on CT cannot be performed, HU are utilized to predict mild, moderate, and severe steatosis as previously described [23,25,26,27]. These measurements may not be robust enough to capture all patients with >5% steatosis, and thus some patients may have been mislabeled as “No NAFLD”, however, we expect that using CT is more accurate in diagnosing NAFLD than noninvasive scores such as the FLI or using medical records or Medicare claims data. There is also the potential for misclassification of dietary intake with self-reported diet. Furthermore, we cannot make any comment regarding the types of animal products that may be associated with NAFLD given that meat is grouped together. Lean meats are likely less associated than those high in saturated fat. Due to the cross-sectional nature of this study, results should be interpreted as describing observed associations and not causal relationships. Finally, the results of this study may not be generalizable beyond Black and White middle-aged adults in the United States. We expect that the associations between APDQS and NAFLD may be generalizable to adults living outside of the US, i.e., not population dependent since foods characterized as beneficial in the US would also presumably be beneficial in other countries. However, the primary dietary patterns of non-US populations likely differ from the patterns derived in CARDIA, and thus these results may not be comparable. 

In conclusion, associations between NAFLD and both the APDQS and meat dietary patterns in CARDIA participants are entwined with the association of these dietary patterns with higher generalized obesity. A follow-up from the year 30 CARDIA exam may provide additional insight into these questions.

## Figures and Tables

**Figure 1 nutrients-14-00854-f001:**
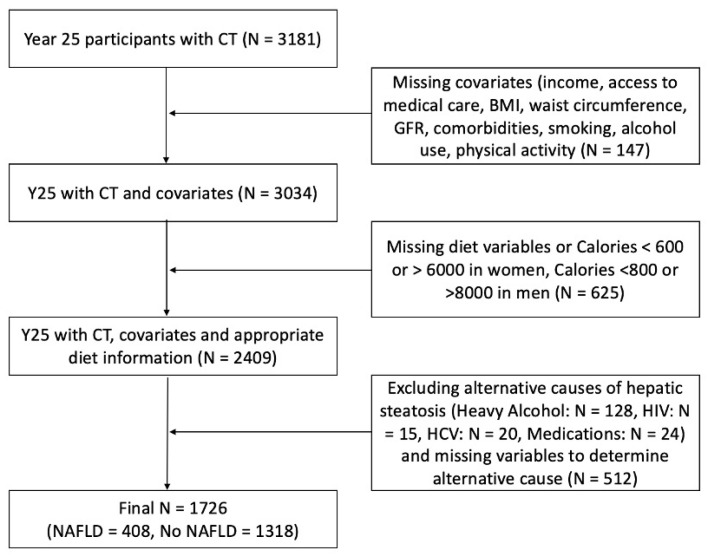
Flowchart of participants at Year 25 included in final analysis. GFR, glomerular filtration rate; HIV, human immunodeficiency virus; HCV, chronic hepatitis C.

**Figure 2 nutrients-14-00854-f002:**
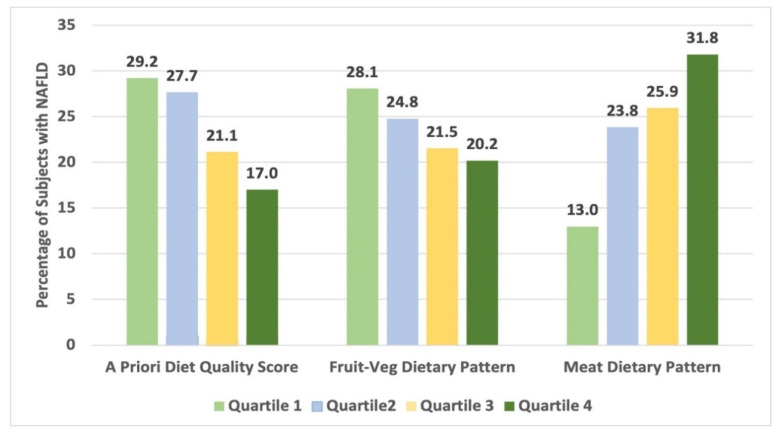
NAFLD proportions by diet quality score and dietary patterns.

**Table 1 nutrients-14-00854-t001:** Demographics and clinical characteristics of patients with and without NAFLD.

	No NAFLD *N* = 1318	NAFLD ^1,2^ *N* = 408	*p*-Value
Age (year), mean (SD)	50.1 (3.6)	50.3 (3.6)	0.25
Women (%)	813 (61.7)	185 (45.3)	<0.0001
White (%)	780 (59.2)	251 (61.5)	0.40
Socioeconomic status			
Highest grade completed, mean (SD)	16.04 (2.5)	15.72 (2.5)	0.018
Income > $50,000/year (%)	950 (72.1)	279 (68.4)	0.15
Access to medical care			
Report regular medical care (%)	1215 (92.2)	383 (93.9)	0.26
Difficulty accessing healthcare (%) ^3^	133 (10.1)	36 (8.8)	0.45
BMI (kg/m^2^), mean (SD)	28.2 (6.0)	34.2 (6.9)	<0.0001
Obese, BMI > 30 (%)	394 (29.9)	292 (71.6)	<0.0001
Waist circumference (cm), mean (SD)	89.1(13.3)	106.4 (14.0)	<0.0001
Glomerular filtration rate (mL/min/1.73 m^2^), mean (SD)	93.9(19.4)	96.7 (21.3)	0.020
Comorbidities (%)			
Hyperlipidemia ^4^	225 (17.1)	186 (45.6)	<0.0001
Hypertension ^5^	401 (30.4)	199 (48.8)	<0.0001
Diabetes mellitus ^6^	101 (7.7)	110 (27.0)	<0.0001
Metabolic syndrome ^7^	129 (9.8)	181 (44.4)	<0.0001
Alcohol use (g/day), median (IQR)	5.5 (15.2)	5.1 (15.0)	0.38
Smoking (%)	175 (13.3)	73 (17.9)	0.020
Physical activity (exercise units/week), median (IQR)	305 (369)	270 (335.5)	0.013

NAFLD group includes those with mild steatosis (^1^ Liver attenuation > 40 and ≤51 Hounsfield units) and moderate-severe steatosis (^2^ Liver attenuation ≤40 Hounsfield units). ^3^ Responded “hard” or “very hard” in response to survey question “How hard is it to get needed health services?”. ^4^ Total cholesterol ≥240 mg/dL and/or lipid-lowering therapy. ^5^ Antihypertensive medication use and/or systolic blood pressure ≥140 mmHg or diastolic blood pressure ≥90 mmHg. ^6^ Fasting plasma glucose ≥126 mg/dL, treatment with insulin or hypoglycemic agent, 2-h post-challenge glucose ≥200 mg/dL and/or hemoglobin A1C ≥6.5%. ^7^ Defined using Adult Treatment Panel III criteria; SD: Standard Deviation. *p*-values from Chi-squared or *t*-tests/Wilcoxon.

**Table 2 nutrients-14-00854-t002:** Association of intake and food groups with and without NAFLD.

	No NAFLD *N* = 1318	NAFLD ^1,2^ *N* = 408	*p*-Value
Energy intake (kcal/day)	2276 (985)	2466 (1111)	0.002
Food groups (servings/day)			
Fruit	2.70 (2.31)	2.42 (2.20)	0.031
Vegetables (excluding potatoes)	3.82 (2.65)	3.72 (2.73)	0.48
Vegetables, potatoes	0.43 (0.44)	0.54 (0.55)	0.0003
Grains	6.41 (3.58)	7.01 (3.69)	0.004
Meat and fish	5.12 (3.42)	6.05 (3.90)	<0.0001
Dairy	2.47 (2.78)	2.76 (2.51)	0.048
Fats	5.22 (5.74)	6.14 (8.09)	0.032
Beans	0.24 (0.39)	0.24 (0.35)	0.96
Eggs/omelets	0.55 (0.67)	0.61 (0.55)	0.089
Seeds, nuts, peanut butter	1.21 (2.01)	1.04 (1.61)	0.077
Salad dressings/sauces	2.34 (2.02)	2.33 (1.80)	0.86
Soups	0.05 (0.09)	0.06 (0.09)	0.10
Soy/nondairy products	0.73 (1.51)	0.71 (1.78)	0.90
Pickled foods	0.50 (1.51)	0.43 (0.64)	0.13
Chocolate	0.20 (0.45)	0.19 (0.38)	0.74
Sweet extras	1.81 (4.00)	1.50 (2.56)	0.068
Sugar substitutes	0.66 (2.31)	0.62 (1.96)	0.69
Beverages	4.48 (3.26)	4.98 (3.52)	0.009

NAFLD group includes those with mild steatosis (^1^ Liver attenuation > 40 and ≤51 Hounsfield units) and moderate-severe steatosis (^2^ Liver attenuation ≤ 40 Hounsfield units). Mean (SD) provided. *p*-values from *t*-test, unadjusted.

**Table 3 nutrients-14-00854-t003:** NAFLD at year 25 as a function of a priori diet-quality score and meat and fruit-vegetable dietary patterns at year 20 in 3 separate logistic regression models.

Dietary Score/Pattern	NAFLD ^1^ *N* = 408 OR (CI) (Adjusted Model 1) *	*p*-Value (Adjusted Model 1) *	NAFLD ^1^ *N* = 408 OR (CI) (Adjusted Model 2) **	*p*-Value (Adjusted Model 2) **	NAFLD ^1^ *N* = 408 OR (CI) (Adjusted Model 3) ***	*p*-Value (Adjusted Model 3) ***
*A* priori diet-quality score		0.004		0.040		0.12

Quartile 2 vs. 1	0.95 (0.69, 1.30)		1.14 (0.8, 1.63)		1.03 (0.70, 1.50)	
Quartile 3 vs. 1	0.67 (0.47, 0.95)		0.79 (0.54, 1.17)		0.80 (0.53, 1.21)	
Quartile 4 vs. 1	0.53 (0.36, 0.79)		0.66 (0.43, 1.01)		0.63 (0.40, 1.00)	
Meat dietary pattern		<0.0001		0.014		0.27
Quartile 2 vs. 1	1.93 (1.34, 2.79)		1.56 (1.05, 2.33)		1.38 (0.91, 2.09)	
Quartile 3 vs. 1	2.08 (1.43, 3.02)		1.55 (1.04, 2.32)		1.34 (0.88, 2.04)	
Quartile 4 vs. 1	2.70 (1.83, 3.99)		2.01 (1.32, 3.05)		1.55 (1.00, 2.42)	
Fruit-vegetable dietary pattern		0.40		0.46		0.15
Quartile 2 vs. 1	0.86 (0.62, 1.19)		0.85 (0.6, 1.23)		0.83 (0.56, 1.21)	
Quartile 3 vs. 1	0.80 (0.56, 1.13)		0.86 (0.59, 1.27)		0.78 (0.52, 1.18)	
Quartile 4 vs. 1	0.74 (0.51, 1.06)		0.72 (0.48, 1.08)		0.60 (0.39, 0.93)	

^1^ Liver attenuation ≤ 51 Hounsfield units. Groups-A Priori Score: 0 = 29–54, 1 = 55–63, 2 = 64–72, 3 = 73–99; Meat dietary pattern: 0 = −1.6904–−0.7020, 1 = −0.7017–−0.2532, 2 = −0.2519–0.4378, 3 = 0.4415–5.4131; Fruit-vegetable pattern: 0 = −2.153–−0.618, 1 = −0.6179–−0.1104, 2 = −0.1101–0.5151, 3 = 0.5161–6.053. * Adjusted model 1 (adjusted for age, race, sex, education, alcohol use, physical activity, center, smoking at Year 25), type 3 *p*-value. ** Adjusted model 2 (adjusted for age, race, sex, education, alcohol use, physical activity, center, smoking, and comorbidities (type 2 diabetes mellitus, dyslipidemia, hypertension, metabolic syndrome) at Year 25), type 3 *p*-value. *** Adjusted model 3 (adjusted for age, race, sex, education, alcohol use, physical activity, center, smoking, BMI, comorbidities (type 2 diabetes mellitus, dyslipidemia, hypertension, metabolic syndrome) at Year 25), type 3 *p*-value.

## Data Availability

The data presented in this study are openly available in https://biolincc.nhlbi.nih.gov/studies/cardia/?q=CARDIA (accessed on 7 January 2022).

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
