# Peer review of "Dietary Patterns and Prevalent NAFLD at Year 25 from the Coronary Artery Risk Development in Young Adults (CARDIA) Study"

_nutrients, 2022, doi:10.3390/nu14040854_

Round 1

Reviewer 1 Report

Minor comment:
Page 9. Lines 282-283: The sentence that the findings of the present study differ from prior analyses which have shown that a poorer diet quality is associated with an increased risk of NAFLD reads a bit odd and seems somewhat exaggerated as the data from the logistic regression models 1 and 2 go into the same direction; only with model 3 which also included  BMI the statistically significant findings were diminished.

Author Response

  1. Page 9. Lines 282-283: The sentence that the findings of the present study differ from prior analyses which have shown that a poorer diet quality is associated with an increased risk of NAFLD reads a bit odd and seems somewhat exaggerated as the data from the logistic regression models 1 and 2 go into the same direction; only with model 3 which also included BMI the statistically significant findings were diminished.

Response: The authors have revised this sentence to read “Our findings are similar to prior analyses…”.

Reviewer 2 Report

The manuscript submitted for publication to Nutrients by Gray et al., titled: "Dietary patterns and prevalent NAFLD at year 25 from the Coronary Artery Risk Development in Young Adults (CARDIA) study" is a follow-up study from a population sample of the CARDIA study investigating dietary associations with NAFLD prevalence. 

The manuscript is well-written addressing an interesting and important topic with clinical and public health implications. The manuscript is organized and structured well, flows logically and is easy for the reader to follow.

The reviewer would like to offer a few points below for consideration by the authors for the improvement of the manuscript:

  1. While the CARDIA study presents the inclusion and exclusion criteria and the rationale for participant selection it would be useful to include some of those parameters as well as (more importantly) how confounding factors were identified and addressed/normalized for in the reported study.
  2. BMI does not have units. It is an index and is calculated by the division of weight in kg over the height in m squared but this calculation does not confer units. Unfortunately a mistake that seems gradually more common even in published literature.
  3. Consider including a hypothesis at the end of the introduction section.
  4. In the discussion section it would be useful to discuss how results may be different or similar per setting if compared to other countries.
  5. How applicable are those conclusions outside of the US?
  6. What do the authors believe is/are the strongest drivers for the obtained results and why?

Author Response

  1. While the CARDIA study presents the inclusion and exclusion criteria and the rationale for participant selection it would be useful to include some of those parameters as well as (more importantly) how confounding factors were identified and addressed/normalized for in the reported study.

Response: The authors added the following sentence (line 94-96). “Possible confounding variables, including demographic, clinical, and others, were selected based on literature review and known clinical risk factors. These variables were adjusted for in the multiple logistic regression models.”

  1. BMI does not have units. It is an index and is calculated by the division of weight in kg over the height in m squared but this calculation does not confer units. Unfortunately, a mistake that seems gradually more common even in published literature.

Response: We respectfully disagree with the reviewer that BMI does not have units. We believe that the units “kg/m2” are appropriate and widely accepted. However, we will abide by the editor’s recommendation on this issue.

  1. Consider including a hypothesis at the end of the introduction section.

Response: The authors note that a hypothesis is stated at the end of the introduction (lines 71-74) already and thus no additional changes were made to address this comment.

  1. In the discussion section it would be useful to discuss how results may be different or similar per setting if compared to other countries.

Response: The authors added the following sentence to the discussion section (line 885-890). “Finally, the results of this study may not be generalizable beyond Black and White middle-aged adults in the United States. We expect that the associations between APDQS and NAFLD may be generalizable to adults living outside of the US, i.e., not population dependent since foods characterized as beneficial in the US would also presumably be beneficial in other countries. However, the primary dietary patterns of non-US populations likely differ from the patterns derived in CARDIA, and thus these results may not be comparable.”

  1. How applicable are those conclusions outside of the US?

Response: The authors added an additional sentence in the discussion section (lines 885-890) to address how the results may or may not be applicable outside of the US.

  1. What do the authors believe is/are the strongest drivers for the obtained results and why?

Response: The authors added the following sentences (line 837-842) to address this comment. “The APDQS is a useful score for diet quality in NAFLD as the classification of foods as beneficial, neutral, and adverse is in line with current data with regards to beneficial (fruits, vegetables, whole grains, lean meats, and unsaturated fats) and adverse (refined carbohydrates and saturated fats) foods for NAFLD.[3] Similarly, the meat dietary pattern would be expected to represent a diet high in saturated fat, known to be harmful in NAFLD.”

Round 2

Reviewer 2 Report

The authors have reasonably addressed reviewer's comments.